# Vision-Based HAR in UAV Videos Using Histograms and Deep Learning Techniques

**DOI:** 10.3390/s23052569

**Published:** 2023-02-25

**Authors:** Sireesha Gundu, Hussain Syed

**Affiliations:** School of Computer Science and Engineering, VIT-AP University, Amaravati 522237, India

**Keywords:** activity recognition, Bi-LSTM, deep learning techniques, HOG, instance segmentation, Mask-RCNN

## Abstract

Activity recognition in unmanned aerial vehicle (UAV) surveillance is addressed in various computer vision applications such as image retrieval, pose estimation, object detection, object detection in videos, object detection in still images, object detection in video frames, face recognition, and video action recognition. In the UAV-based surveillance technology, video segments captured from aerial vehicles make it challenging to recognize and distinguish human behavior. In this research, to recognize a single and multi-human activity using aerial data, a hybrid model of histogram of oriented gradient (HOG), mask-regional convolutional neural network (Mask-RCNN), and bidirectional long short-term memory (Bi-LSTM) is employed. The HOG algorithm extracts patterns, Mask-RCNN extracts feature maps from the raw aerial image data, and the Bi-LSTM network exploits the temporal relationship between the frames for the underlying action in the scene. This Bi-LSTM network reduces the error rate to the greatest extent due to its bidirectional process. This novel architecture generates enhanced segmentation by utilizing the histogram gradient-based instance segmentation and improves the accuracy of classifying human activities using the Bi-LSTM approach. Experimental outcomes demonstrate that the proposed model outperforms the other state-of-the-art models and has achieved 99.25% accuracy on the YouTube-Aerial dataset.

## 1. Introduction

Drone technology has advanced considerably in recent years. Drones are becoming more and more useful in places a man cannot quickly and effectively reach. When drone technology is implemented in various fields such as industries, government agencies, military sites, and so on, the scope, potential, and scale of global reach increase. They can reach the most remote places, where little manpower, effort, energy, or time is required.

The next generation of drones will primarily focus on propulsion, size, and autonomy. The type of drone is determined by the technology employed to fly it [1]. Multirotor drones are the most common type of drone, and most professionals use them. Video surveillance, aerial photography, and other multirotor drone applications are just a few examples. A multirotor camera allows for more precise framing and positioning of the camera, resulting in crisp aerial photos. Multirotors are the most cost-effective to fly and construct. Quadcopters are the most extensively used and popular of the numerous forms of multirotor. Multirotors, however, have several drawbacks. They have a finite quantity of endurance, speed, and flight time. The maximum flight time for a multirotor is 20–30 min with a minimal payload, and these are the most common.

Interpreting the articulation of a human physique in an image and detecting the person’s motion in a video sequence acquired by multirotor quadcopter drones [2] is a difficult research topic. It can be difficult to discern human activities in a variety of situations, such as visual blur, perspective distortion, low-resolution scenes, occlusions, and so on.

Many effective research efforts in human activity recognition have been carried out using various deep learning approaches using numerous common video datasets. Because of the endless postures and articulated character of the human being, human activity recognition is a complex and time-consuming study challenge. Researchers are increasingly interested in human activities to develop autonomous driving systems that use a variety of data collection methods. This necessitates examining images and videos captured by aerial cameras [3].

Many datasets are available on the internet to research the behavior of people, automobiles, and the road environment. These datasets are crucial for the study of autonomous vehicle systems, including buses and self-driving vehicles. Numerous academics are now working on real-time research projects and commercial projects in fields such as search and rescue, crowd control, situational awareness, surveillance, and sports activity recording [4,5,6].

To perform at a high level in these challenging environments, one needs efficient and suitable algorithms as well as training data. The challenging missions need datasets containing a variety of human perspectives. The vast majority of datasets focus on identifying aerial or ground-based activity. Finding a dataset containing video sequences of ground-level and aerial-view-based human activities is difficult.

The Microsoft Common Objects in Context (MS-COCO) dataset [7], which includes numerous views and footage of one or more individuals, was utilized to research human identification, human activity recognition, and other topics. The majority of the activities in this dataset have different perspectives, which are known as aerial platforms, while the activities from the ground-level sights are photographed by a rigid or flying camera. This paper mainly focuses on single and multiple human activity recognition by utilizing spatial and temporal features. To improve the accuracy and speed of human recognition, it uses feature descriptor followed by instance segmentation and bidirectional long short term memory (Bi-LSTM) for activity recognition.

The major key contributions of the present research work can be summarized as follows:After dividing the video stream into several frames, the preprocessing pipeline is used to increase the classification efficiency; even more, here rectangular regions of interest (ROIs) are produced based on Sobel edge detection resulting in faster processing as the research is interested in persons and their behavior.Following that, the HOG descriptor is utilized to extract the features from the preprocessed frames to enhance the performance of the model. In this, the intensity distribution of gradients or the direction of contours can describe the local appearance and shape of an object in an image. These descriptors can be implemented by dividing the image into small connected regions called cells. Then, for each cell, a histogram of gradient directions or edge orientations for all pixels in the cell is computed. The descriptor is the sum of these histograms.Next, the extracted HOG features are then communicated to the Mask-RCNN framework, which is pretrained on the MS-COCO model, to improve prediction accuracy and achieve cutting-edge outcomes for human detection.Finally, Bi-LSTMs, as opposed to baseline LSTMs (which use only past information), use both past and future information when the entire sequence of time series data is available. Because of the additional context provided, the network can make more accurate predictions. The model employs convolutional kernels of various sizes, which allows it to capture various temporal local dependencies in sequential data.

Some of the limitations of the previous work are described as follows: the temporal dimension and global optimization techniques have not been incorporated into the mask-based object segmentation and tracking that are suggested in [8]. In order to modify the FCN based on temporal input, the model in [9] was unable to learn a recurrent representation of the modulation parameters. The object proposal and detection with spatial temporal features, end-to-end trainable matching criterion, and including motion information for better recognition and identity association were not used in another video instance segmentation model [10]. Human group detection and tracking for event detection utilizing the action recognition paradigm for HOG [11] was unsuccessful. The accuracy and performance of the suggested model for real-time research applications in deep learning-based object detection using Mask-RCNN [12] failed to improve. Robust pedestrian detection using a recursive convolution neural network [13] and human detection and tracking with deep convolutional neural networks while restricted by noise and obscured scenes [14] failed to fully implement special and temporal features and increasing the accuracy in detecting and tracking humans.

Detecting complex activities becomes time-consuming and difficult as more features are gathered. These methods have limitations on accuracy and call for specialized training and expertise in the relevant field. This is where the proposed deep learning-based HAR with histograms has proved beneficial. Deep learning models such as Mask-RCNN and bi-directional LSTM automatically learn the features required to make accurate predictions from the histogram data which is taken from raw data directly. This enables new and large datasets to be used for HAR. Drone-captured YouTube-Aerial data is used, which results in an efficient model. This model is also capable of learning high-level features which can be very well utilized in complex HAR.

The following sections make up this paper within the field of image processing. The relevant research on object detection and segmentation, human detection, histogram of oriented gradient (HOG), instance segmentation techniques, regional convolutional neural network (RCNN), and activity recognition in videos is listed in Section 2. The problem definition is described in Section 3. The proposed methodology is presented in Section 4. Section 5 and Section 6 provide descriptions of the HOG and the mask-regional convolutional neural network (Mask-RCNN). Section 7 discusses the long short-term memory (LSTM) and Bi-LSTM architectures. Section 8 describes the dataset, the metrics that were employed, the experimental findings, and comparisons to earlier models. The work’s conclusion is presented in Section 9.

## 2. Literature Survey

Using Microsoft Kinect, Stone et al. [15] employed an adult and a two-stage fall detection model for senior citizens. When viewed from the ground, a person’s vertical position is first identified using individual depth frames, and in the following phase, the processing is carried out by employing time-series segmentation of the vertical position of the human from the ground inputs. This approach is especially beneficial for older persons, as it produces superior results in actions such as standing, lying down, and sitting. When compared to conventional fall detection mechanisms, it produced better results.

In order to handle complicated behaviors and group dynamics in streaming sequences, Zhuang et al. [16] proposed a method that combines differential recurrent convolutional neural networks (DRCNN) and stacked differential long short-term memory (DLSTM).

Human motions and actions are essential for the understanding of video analysis and human activities, according to Cheng et al. [17]. The suggested model focuses on interactions between humans, with a restricted number of individuals cooperating to achieve a common objective. The motion trajectories in this model were based on the Gaussian model. It enhances recognition accuracy.

Social signal processing (SSP) is a novel perspective on mechanized human action surveillance that combines some psychological principles that are both effective and social, according to Cristani et al. [18]. This model offered nonverbal clues, which are typically used in conscious aware systems, such as body gestures and posture, facial expression, vocal aspects, and gaze.

By leveraging the Kinect sensor, Yoon et al. [19] developed a procedure for computer vision applications using the Kinect that overcame fundamental computer vision problems. This method consists of preprocessing, object tracking and recognition, analysis of human activity, indoor 3D mapping, and analysis of hand gestures.

In order to simultaneously distinguish different movements and actions, a prototype was developed using a gym as an example, according to Ling et al. [20]. This prototype also includes color intensity-based segmentation of human gestures into temporal sequences and motion-based algorithms for efficient human action segmentation. Human action’s shape and features will be apparent.

An optical flow-based model approach to identify activity in the footage was put forth by Shao et al. [21]. Using the optical flow approach, the monitored point of interest is sorted using the k-means method into several clusters. Each cluster projection’s displacements are crucial in determining the direction, geometric location, and principal component of each cluster. These estimates reveal the highest likelihood of a high-activity cluster encompassing video events.

Object segmentation in videos: there are two methods for segmenting video objects: unsupervised and semi-supervised. Semi-supervised segmentation of video objects [22] emphasizes mask-based object segmentation and tracking. Temporal consistency, motion cues, and visual similarity are collected from the video to locate the related object [23,24,25]. Segmentation is applied to a single foreground item in an unsupervised setting [26,27,28]. The suggested algorithms ignore semantic categories in both circumstances and treat the target items as general objects. Using instance segmentation to recognize objects in videos is also becoming more popular.

Object detection in videos: detecting objects in video streams is referred to as video object detection. Initially, it is a visual challenge from ImageNet [29]. Object identity information is frequently employed to improve the robustness of detection methods [30,31,32]. Object detection and tracking are not required for the evaluation metric, which is confined to per-frame detection.

Human detection using HOG: the research field of object detection is wide, however, we just list a few pertinent articles on person detection here. A polynomial support vector machine (SVM) based person detection using rectified Haar wavelets as input descriptors are described by O Pinheiro et al. [33], with a part (sub-window)-based variant in [34]. Dalal and Triggs [35] adopt a far more forthright technique, retrieving edge pictures and utilizing chamfer distance to compare them to a collection of accomplished exemplars. Dai et al. [36] employed AdaBoost to train a series of increasingly complex region rejection rules based on space-time disparities and Haar-like wavelets, resulting in an efficient moving person detector. Dai et al. [37] developed a parts-based technique with detectors for heads, faces, and front and side profiles of lower and upper body parts, combining binary-thresholded gradient magnitudes and orientation position histograms.

Instance segmentation: this separates pixels into semantic classes, after which it creates instances of objects [38]. It employed a two-stage approach that combines the usage of region proposal network (RPN) to generate object proposals with the aggregate of region of Interest (RoI) features to anticipate object classes, bounding boxes, and masks [39,40]. Many techniques, such as RCNN, use segment proposals to implement this approach. Bottom-up segments were employed in the previous approach [41]. Subsequent works [42,43] offered segment candidates, which fast R-CNN classified. Dai et al. [44] proposed a method that uses a sophisticated multi-stage cascade to predict segment proposals from bounding-box proposals, with classification as the final step. Ren et al. [45] recently proposed a prototype for “Fully Convolutional Instance Segmentation (FCIS)” that integrated an object identification system and a segment proposal system. The simple principle [46] is to forecast completely convolutional output channels that are location sensitive. Our detector, on the other hand, has a fundamental structural design with only one detection window, but it seems to perform substantially better on pedestrian imagery.

RCNN: the region-based convolutional neural network (R-CNN) architecture, described in [47], is used to determine the bounding box of an object and to handle a large number of potential object regions, as well as to evaluate convolutional networks separately on each RoI [48,49]. Region of interest pooling (RoIPool) is used by R-CNN to swiftly, accurately, and efficiently build RoIs on feature maps [50]. For more reliable and adaptable subsequent enhancements, R-CNN employs an RPN with an attention mechanism.

Activity recognition in videos: recurrent neural Network (RNN) is a very effective and extensively used network architecture in sequential modeling applications such as human activity recognition in videos. The LSTM is an RNN-based network that is widely used for learning motion properties in video-based activity recognition [51]. This can also be leveraged to mitigate gradient expansion and gradient vanishing difficulties during the training phase to some extent. Based on LSTM architecture, another study proposed a video as an ordered sequence using bidirectional LSTM architecture [52]. Bi-LSTMs outperform unidirectional LSTMs in terms of prediction. Bi-LSTM architecture has been used in numerous video-related applications, such as video-super resolution, object segmentation in the video, spatiotemporal feature learning for gesture identification, and fine-grained action detection. Long-term dependencies are well-handled by Bi-LSTMs. Unlike LSTMs (which only use past data), Bi-LSTMs employ both past and future data when the entire time-series sequence data is available, allowing the network to generate more accurate predictions. Following that, bidirectional LSTMs were used to predict frame-wise phoneme categorization, network-wide traffic speed, and other variables [53]. Only a few research papers in the field of activity recognition make proper use of the Bi-LSTM network. We present a novel model that uses HOG with Mask-RCNN architecture for edge detection and segmentation of humans in images, as well as Bi-LSTM architectures for learning spatiotemporal aspects of neighboring video frames.

## 3. Problem Definition

Human activity recognition has received a fair amount of study attention in static camera-based surveillance and is a relatively well-researched topic, whereas human activity recognition in unmanned aerial vehicle (UAV) captured aerial videos is comparatively understudied. It has received much attention in recent years owing to open-source aerial videos available on social media. It is a difficult topic to solve because human activity recognition can not be anticipated efficiently from aerial videos. Another complex problem is detecting single and multiple-human activity in UAV videos when there is background clutter, occlusion, background lighting change, loss of spatiotemporal features, substantial intra-class variance among specific classes and low inter-class variance, image distortion, and a person pose variation. This paper introduced the novel approach to detecting and recognizing single and multiple human actions by employing spatial and temporal features. Experimental outcomes show that the proposed methodology outperforms the existing models.

## 4. Proposed Methodology

The proposed method is shown in Figure 1, which takes a video stream as input and splits it into several frames. To reduce training and detection time, the preprocessing pipeline method is used. The rectangular region of interest is produced by this technique, which is based on Sobel edge detection [54]. A smaller zone of interest is included in this extracted rectangular portion, which means there are fewer pixels to process, resulting in faster processing. Because we are interested in persons and their behavior, this enables us to apply a more straightforward preprocessing method that merely extracts the RoIs.

A multirotor quadcopter-equipped drone can employ the precompiled Mask-RCNN with HOG features and can be used by a multirotor quadcopter-mounted drone for person detection. Mask-RCNN with HOG features for person detection to capture human images. The next stage is to extract features with HOG [55]. This stage extracts an object’s features, which are subsequently communicated to the Mask-RCNN framework in order to increase prediction accuracy.

On established benchmarks, the suggested approach utilizes the Mask-RCNN network to produce cutting-edge results for human detection. For object detection challenges, this architecture was trained using the MS-COCO model. Mask-RCNN’s performance in the area of image processing seldom reached similar outcomes due to the complicated kind of aerial imagery whose characteristics are acquired by the HOG technique, diverse object scales, and a pool of annotated data. This study investigates object region proposal creation, pixel-based segmentation, alignment of RoI, bounding box regression, and classification to recognize the human in UAV recordings. SoftMax classifier is used to classify people among various objects, and RoIPool is used to derive features from the bounding boxes.

## 5. HOG Descriptor

The field of applied machine learning is referred to as feature engineering which involves the extraction of additional features from existing raw data to enhance the performance of the model. Histogram of oriented gradients (HOG) is the term for an antiquated technique for feature extraction. The following sections will go over the foundations and functionality of the HOG feature representation.

The following are the guiding design principles for computer vision features:Interpretation of feature descriptorPrinciples of HOGProcess of HOG calculation:
*Data preprocessing*Evaluation of magnitudes*Evaluation of magnitude and direction*Evaluation of histogram of magnitudes
☞Construct histograms with magnitudes and orientations*Normalization of magnitudes*Produce HOG features for a complete image.

### 5.1. Interpretation of Feature Descriptor

A feature descriptor is a concise summary of a frame that only has the data essential to identifying its objects (such as the shape of the object, color, edge, backdrop, and so on). HOG is the most often used feature descriptor algorithm (together with HOG, scale-invariant feature transform (SIFT), sped-up robust features (SURF), and others).

### 5.2. Principles of HOG

A general computer vision task for object detection is the HOG feature descriptor, which identifies patterns in picture data and extracts them.

The following are some ways that the HOG is unique compared to other feature descriptors: An object’s main priorities are its shape and structure. The magnitude and direction (or gradient and orientation) of the edges are extracted by HOG in order to determine whether a pixel serves as both an edge and a direction for edges. The directions are established in the specific regions of a frame. This reveals that the frame is fragmented into a large number of smaller regions. The magnitude and direction of each of these zones are analyzed.

Then, all of these regions are divided by HOG into a unique histogram. A “Histogram of Oriented Gradient” is a histogram that is produced utilizing the magnitude and the direction of pixel values.

Finally, the fundamental principle of HOG is that it keeps note of when a gradient orientation occurs in particular localized regions of a frame.

### 5.3. Hog Calculation

The input frame for identifying the HOG features with a resolution of 298 × 169 pixels is shown in Figure 2.

#### 5.3.1. Data Preprocessing (64 × 128)

Preprocessing data is a key stage in most machine learning studies and when working with images. HOG preprocessing maintains a fixed, uniform aspect ratio for each image patch regardless of image size. In our case, the patches must have a 1:2 aspect ratio. They can, for example, be 200 × 400, 256 × 512, or 1000 × 2000, but not 106 × 220. In order to extract the features, and make calculations easier, Figure 3 shows how the frame is split into 8 × 8 and 16 × 16 patches in HOG with a 1:2 width-to-height ratio. They are based on the input image size and the output feature vector length. Patches at various scales are typically analyzed and tested at multiple image locations. The only limitation is that the patches under consideration have a fixed aspect ratio.

#### 5.3.2. Evaluation of Magnitudes (X and Y Direction)

In this stage, determine the size of the small orientation shifts in the X and Y axes for every individual pixel in the frame. Presume the magnitude of a small portion of an image such as the one in Figure 4a. The pixel matrix shown in Figure 4b is a matrix that depicts the pixel values of the chosen patch.

The directional change (gradient/magnitude) for the highlighted pixel value 95 will now be computed for both the X and Y axes. Subtract the value of the left pixel from the value of the right pixel to determine a single pixel’s magnitude in the X-direction. Subtract the value of the bottom pixel from the value of the top pixel to obtain the magnitude in the Y-direction.

Hence, Magnitude in X-direction ($M_{x}$) = 90 − 82 = 8; Magnitude in Y-direction ($M_{y}$) = 68 − 62 = 6.

These two metrics are used in order to save magnitudes in the X and Y directions, respectively. The size 1 Sobel kernel filter is implemented using the same methodology. Repeat the process for all of the pixels in the image. The difference in intensity along the edges is particularly sharp, resulting in a larger magnitude. The magnitude and orientation of the object are then determined using these measurements.

#### 5.3.3. Evaluation of Magnitude and Direction

Use the Pythagoras theorem to determine the magnitude and orientation of each pixel value. Consider the right-angle triangle shown in Figure 5.

In this figure, the gradients $M_{x}$ and $M_{y}$ (8 and 6 in our case) are base and perpendicular. According to Pythagoras theorem, the following Equation (Equation 1) is used to calculate the total gradient magnitude:(1)$$\mu =|\sqrt{M_{x}^{2}+M_{y}^{2}}|$$

Hence, the total magnitude of the gradient is $|\sqrt{8^{2}+6^{2}}|=10$.

The direction (or orientation) of the same pixel must now be calculated. To do so, the following Equation (Equation 2) is used to determine angles with a tan:(2)$$\theta =|\tan^{-1}\left(M_{y}/M_{x}\right)|$$

The orientation value is 36.88 (≈37) when the aforementioned values are given in the computation. This method allows us to determine each pixel’s gradient and direction, and these gradients and directions can be utilized to build the histograms in the following step.

#### 5.3.4. Evaluation of Histogram of Magnitudes in 8 × 8 Cells (9 × 1)

It is important to comprehend what a histogram is and how to make one using magnitudes and orientations before we can compute the magnitudes needed to generate them.


1.Construct histograms with magnitudes and orientations:


A histogram is a graphical illustration of how frequently each bin occurs for a given set of continuous data. We use this method to determine the orientation of each pixel and log the occurrence of these values in a 9 × 1 matrix, as shown in Figure 6 bins. We use a bin size of 20 and a bucket count of 9.

The image’s histogram must then be created as the next step. As illustrated in Figure 7 cells, partition the entire image into 8 × 8 cells, and then compute HOG for each cell. As a result, each cell receives a 9 × 1 matrix in addition to the histograms for every smaller patch of the overall image. For instance, its value can be modified to 16 × 16 or 32 × 32 from 8 × 8 or vice versa. After this stage, the histograms must then be normalized.

#### 5.3.5. Normalization of Magnitudes for a 16 × 16 Cell (36 × 1)

For normalizing each block, Triggs and Dalal provided four potential strategies. Assume that ||v|| is the n-norm for n={1,2}, *v* is a non-normalized vector containing all of a block’s histograms, and *a* is a small constant added to the square of ||v|| to prevent zero division error. The normalization factors are calculated using the following Equation (Equation 3):(3)$$L2-norm:g=\frac{v}{\sqrt{{\left||v|\right|}_{2}^{2}+a^{2}}}$$

L2-hys: L2-norm is clipped and renormalized afterward. In this instance, restrict the maximum values of *v* to 0.2.
(4)$$L1-norm:g=\frac{v}{\left(||v|{|}_{1}+a\right)}$$
(5)$$L1-sqrt:g=\sqrt{\frac{v}{\left(||v|{|}_{1}+a\right)}}$$
when compared to non-normalized data, the four approaches discussed above perform significantly better. By measuring the L2-norm, clipping the outcome, and then re-normalizing, L2-hys can be produced. According to Triggs and Dalal, the efficiency of the schemes L2-norm, L1-sqrt (stated in Equation (Equation 5)), and L2-hys are comparable, while the performance of the other schemes, L1-norm (stated in Equation (Equation 4)), is noticeably poorer.

The range of pixel intensity values can be changed using a technique called normalization, such as histogram stretching. Normalization is necessary since the magnitudes, for a single image, are sensitive to contrast, brightness, and general illumination. This suggests that while certain areas of a picture are brilliant, others are not. We may not be able to obtain correct histograms as a result of these variances. Though this cannot be eradicated, by using 16 × 16 blocks and gradient normalization, we can greatly reduce the variances in the lighting. The following Figure 8 illustrates how 16 × 16 blocks are produced:

Each 8 × 8 cell produced a 9 × 1 matrix, which was then employed to build a histogram. Joining 8 × 8 cells produced a 16 × 16 block. Therefore, we have the option of using either one 36 × 1 matrix or four 9 × 1 matrices. To normalize this matrix, each of the retrieved values is then divided by the square root of the sum of the squares of these vector values. Take into consideration a vector’s mathematical representation, such as $v=[b_{1},b_{2},b_{3},\ldots ,b_{36}]$.

Now, we use the following Equation (Equation 6) to determine the square root of the sum of the squares of the values in the vector above:(6)$$n=\sqrt{b_{1}^{2}+b_{2}^{2}+\cdots +b_{36}^{2}}$$

Finally, as shown in Equation (Equation 7), divide this number *n* by each of the vector’s u values, and we will obtain the normalized vector with the dimensions 36 × 1.
(7)$$\mathrm{Normalized Vector}=\frac{b_{1}}{n}+\frac{b_{2}}{n}+\ldots \ldots +\frac{b_{36}}{n}$$

#### 5.3.6. Produce HOG Features for Complete Image

The process of developing the histogram features for the entire image is complete at this stage. To generate features for the complete image, we must now merge the 16 × 16 chunks of the single image for which we previously created histogram features. A 64 × 128 image will need 105 (or 7 × 15) 16 × 16 blocks, as seen in Figure 9. There will be a 36 × 1 feature vector per 105 blocks.

As a result, there are 105 × 36 × 1 = 3780 features in total. Now we build HOG features for a picture and check whether they ultimately match the overall number of features. The following Algorithm 1 describes the detailed process of HOG descriptor:
**Algorithm 1** HOG Descriptor**Input:** Aerial videos dataset**Output:** HOG features of all frames**Steps:**1.   Extract frames from each video in the dataset2.   Data preprocessing: Resize all frames to a 1:2 ratio of height and width (i.e., 64 × 128)3.   Calculate the gradients of each pixel in each block of the frame in the X and Y directions   (a)  r,c←rows,columns   (b)    $M_{x}\leftarrow P\left(r,c+1\right)-P\left(r,c-1\right)$   (c)    $M_{y}\leftarrow P\left(r-1,c\right)-P\left(r+1,c\right)$4.  Calculate the magnitude and angle (direction) of each pixel using Equations (Equation 1) and (Equation 2)5.  Divide the gradients matrices into 8 × 8 cells to form a block to calculate a 9-point histogram for each block6.  Let the number of bins and step size be  (a)  Number of bins ←9(between $0^{\circ }$ to $180^{\circ }$)  (b)  Step size $\left(\Delta \theta \right)\leftarrow 180^{\circ }/$ Number of bins (i.e., $20^{\circ })$      For all values in a block calculate the following,      For each $k^{th}$ bin,      i.   The bin boundaries ←[Δθ.k,Δθ.(k+1)]      ii.     Each bin center value be      ii.     $C_{k}\leftarrow \Delta \theta \left(k+0.5\right)$7.  For each cell in the block, calculate $V_{k}$ and $V_{k+1}$ values and append them to the array at the index of $k^{th}$ and ${\left(k+1\right)}^{th}$ bin calculated for each bin  (a)  $k\leftarrow \lfloor \left(\frac{\theta }{\Delta \theta }-\frac{1}{2}\right)\rfloor$  (b)    $V_{k}\leftarrow \mu .\left(\frac{\theta }{\Delta \theta }-\frac{1}{2}\right)$  (c)     $V_{k+1}\leftarrow \mu .\left(\frac{\theta -C_{k}}{\Delta \theta }\right)$8.  Let $v\leftarrow [b_{1},b_{2},b_{3},\ldots ,b_{36}]$   Normalize each block by L2-norm using Equation (Equation 3)9.  Calculate the value of ‘n’ to normalize using Equation (Equation 6) and calculate normalized vector using Equation (Equation 7) where   $g_{n}\leftarrow \left[\left(\frac{b_{1}}{n}\right),\left(\frac{b_{2}}{n}\right),.......,\left(\frac{b_{36}}{n}\right)\right]$


## 6. Mask-RCNN

The implementation of instance segmentation is currently the most difficult task in computer vision. Mask-region convolutional neural network (Mask-RCNN) is a deep neural network architecture that is meant to efficiently incorporate the instance segmentation method to tackle segmentation challenges. It is a two-shot detector that has two stages, the first stage is a region proposal and the second one is a classification of those regions and refinement of the location prediction. In an image or video, it can identify several objects. When an image is provided as input, it outputs object masks, bounding boxes, and classes. It uses a fully convolutional network (FCN) to forecast the mask for each class separately. The Mask-RCNN-based methodology is suitable for this model over the single-shot object detectors such as you only look once (YOLO) frameworks, which are more suitable for real-time localization of objects because the maximum training input image size is 1024 × 1024 whereas YOLO takes 416 × 416. As we use high-resolution images, this architecture helps in the segmentation of humans efficiently compared to other frameworks.

According to [56], object detection based on DCNNs as well as conventional traditional object detection (such as Oxford-MKL [57], DPM [58], NLPR-HOGLBP [59], and selective search [60]) are discussed. It is known that the essential distinction between the two is made by the revival of deep learning, which converts handcrafted object identification features into learned features. High detection accuracy is the primary benefit, and sluggish detection speed is the primary drawback. Examples of two-stage object detection architectures are RCNN [47], SPPNet [50], Fast RCNN [34], Faster RCNN [45], Mask RCNN [40], and RFCN [36]. Others are single-stage object detection designs that use DCNNs to directly locate and classify objects without breaking them up. The class probabilities and location coordinates of an object in a stage can be immediately generated by the one-stage object detection. The region proposal method, which is less complicated than two-stage object detection, is not necessary. The main benefit is quick detection. However, a two-stage object detection design typically provides higher detection accuracy. For instance, one-stage object detection includes OverFeat [61], YOLO series [62,63,64], SSD [65], DSSD [66], FSSD [67], and DSOD [68].

In Mask-RCNN, there are two basic steps of implementation. In the first step, the object bounding boxes are suggested by the region proposal network (RPN) using the input image as a starting point. Based on the first stage’s prediction, the second step determines the object’s class, improves the bounding box, and generates a mask at the pixel level for the object. Both levels are connected by a backbone framework. It is a feature pyramid network (FPN)-style deep neural network. RPN is applied in the three methods listed below.
Bottom-up pathway: it retrieves features from the original frame in a bottom-up fashion. Any convolutional neural network (ConvNet), including visual geometry group network (VGG-net) [69] and residual network (ResNet) [70], can be used.Top-bottom pathway: this leads to a feature pyramid map with the same size as the previous pathway.Lateral connections: these convolutions occur naturally. The primary objective of these connections is to enhance operations between the different levels of the two paths.

RPN, a compact neural network, first assesses all top-down and FPN paths (also termed a feature map). Additionally, it creates areas of interest that include objects. A technique is needed to link newly found features to their raw picture positions when examining the feature map. The scene now includes anchors. Without regard to the image’s content, a set of bounding boxes with predefined scales and locations are called anchors. Individual anchors are assigned bounding boxes based on background binary, intersection over union (IoU) value ground-truth classes, and some IoU value ground-truth classes that are classified in this phase or an object. RPN employs anchors with various scales associated with different feature map layers to locate an object on a feature map and determine the size of its bounding box. To maintain the feature’s positions concerning the object in the original image, convolution, downsampling, and upsampling are used. The algorithmic implementation of Mask-RCNN is represented in the following Algorithm 2.
**Algorithm 2** Procedure for instance segmentation using Mask-RCNN**Input:** Dataset of images with histograms**Output:** Bounding box, mask, class, and score**Steps:**1.   For each image repeat the following steps 2 to 142.   Let upper-left coordinates and lower-right coordinates of predicted and ground truth bounding boxes $B^{a},B^{b}$ be   (a)  $B^{a}\leftarrow \left(x_{1}^{a},y_{1}^{a},x_{2}^{a},y_{2}^{a}\right)$   (b)  $B^{b}\leftarrow \left(x_{1}^{b},y_{1}^{b},x_{2}^{b},y_{2}^{b}\right)$3.   $B^{a}$ requires to meet $x_{2}^{a}>x_{1}^{a}$ and $y_{2}^{a}>y_{1}^{a}:$   (a)  ${\hat{x}}_{1}^{a}\leftarrow min\left(x_{1}^{a},x_{2}^{a}\right)$, ${\hat{x}}_{2}^{a}\leftarrow max\left(x_{1}^{a},x_{2}^{a}\right)$,   (b)  ${\hat{y}}_{1}^{a}\leftarrow min\left(y_{1}^{a},y_{2}^{a}\right)$, ${\hat{y}}_{2}^{a}\leftarrow max\left(y_{1}^{a},y_{2}^{a}\right)$4.   Area of $B^{b}:A^{b}\leftarrow \left(x_{2}^{b}-x_{1}^{b}\right)\times \left(y_{2}^{b}-y_{1}^{b}\right)$5.   Area of $B^{a}:A^{a}\leftarrow \left({\hat{x}}_{2}^{a}-{\hat{x}}_{1}^{a}\right)\times \left({\hat{y}}_{2}^{a}-{\hat{y}}_{1}^{a}\right)$6.   Intersection *I* between $B^{a}$ and $B^{b}:$   (a)  $x_{1}^{I}\leftarrow max\left({\hat{x}}_{1}^{a},x_{1}^{b}\right),x_{2}^{I}\leftarrow min\left({\hat{x}}_{2}^{a},x_{2}^{b}\right)$   (b)  $y_{1}^{I}\leftarrow max\left({\hat{y}}_{1}^{a},y_{1}^{b}\right),y_{2}^{I}\leftarrow min\left({\hat{y}}_{2}^{a},y_{2}^{b}\right)$   (c)    $$I\leftarrow \left\{\begin{array}{cc} \left(x_{2}^{I}-x_{1}^{I}\right)\times \left(y_{2}^{I}-y_{1}^{I}\right), & if\;x_{2}^{I}>x_{1}^{I},y_{2}^{I}>y_{1}^{I}\\ 0, & otherwise \end{array}\right.$$7.    Locating the small enclosed box’s coordinates $B^{c}$:    (a)   $x_{1}^{c}\leftarrow min\left({\hat{x}}_{1}^{a},x_{1}^{b}\right),x_{2}^{c}\leftarrow max\left({\hat{x}}_{2}^{a},x_{2}^{b}\right)$    (b)   $y_{1}^{c}\leftarrow min\left({\hat{y}}_{1}^{a},y_{1}^{b}\right),y_{2}^{c}\leftarrow max\left({\hat{y}}_{2}^{a},y_{2}^{b}\right)$8.    Determine the area of $B^{c}:A^{c}=\left(x_{2}^{c},x_{1}^{c}\right)\times \left(y_{2}^{c},y_{1}^{c}\right)$9.    Determining i and j’s center coordinates    (a)  $C^{a}\leftarrow \left(C_{x}^{a},C_{y}^{a}\right),C^{b}\leftarrow \left(C_{x}^{b},C_{y}^{b}\right)$    (b)  $C_{x}^{a}\leftarrow \left(\frac{{\hat{x}}_{2}^{a}-{\hat{x}}_{1}^{a}}{2}+{\hat{x}}_{1}^{a}\right),C_{y}^{a}\leftarrow \left(\frac{{\hat{y}}_{2}^{a}-{\hat{y}}_{1}^{a}}{2}+{\hat{y}}_{1}^{a}\right)$    (c)  $C_{x}^{b}\leftarrow \left(\frac{x_{2}^{b}-x_{1}^{b}}{2}+x_{1}^{b}\right),C_{y}^{b}\leftarrow \left(\frac{y_{2}^{b}-y_{1}^{b}}{2}+y_{1}^{b}\right)$10.     Calculating the distance between centers:    (a)  $DC\leftarrow \sqrt{{\left(C_{x}^{b}-C_{x}^{a}\right)}^{2}+{\left(C_{y}^{b}-C_{y}^{a}\right)}^{2}}$11.     $IoU\leftarrow \frac{I}{U},$ where $U=A^{a}+A^{b}-I$12.     Perform the non-maximal-suppression Algorithm 3 to choose the highest scoring bounding box13.     Calculate loss function using classification loss, bounding box loss and mask loss by Equations (Equation 8)–(Equation 11)14.     For each RoI, create a mask, class label, bounding box, and score


In the second stage, other neural networks take into account the suggested regions created in the first stage. They propagate to various feature map level areas, scan these areas, and generate multi-category classified object classes, bounding boxes, and masks. This is similar to RPN, except instead of anchors, RoIAlign is used to determine the relative areas of the feature map, and a branch is used to generate masks for each item at the pixel level. The most important feature of Mask-RCNN is the ability to instruct the neural network’s various layers to learn features with different scales, such as RoIAlign and anchors.

### 6.1. Backbone

As a feature extractor, this standard CNN network is used. In this work, each succeeding layer’s low- and high-level features were detected using ResNet101. The image is transformed from 640 × 480 × 3 (RGB) to a 32 × 32 × 2048 shape feature map while the data travels over the backbone network. This serves as the starting point for the subsequent steps.

### 6.2. Feature Pyramid Network (FPN)

Mask-RCNN’s FPN is an extension that can represent things at many scales. To improve conventional feature extraction, It introduces a second pyramid, moving the top-tier features from the previous pyramid to the lower layers that follow. Features at each level can acquire both low-level and high-level features using this approach.

### 6.3. Region Proposal Network (RPN)

In a sliding window analysis, this compact neural network scans the image for regions designated as anchors or boxes that surround it. We choose the top anchors that contain objects based on the forecast and then modify their size and location. The non-max suppression (NMS) technique defined in Algorithm 3 is used to replace the anchors that overlap excessively with the foreground score that is highest and to reject the other anchors. The subsequent stage is then provided with the final RoIs.
**Algorithm 3** A non-maximal suppression Algorithm (NMS)**Input:**   a list of boxes, their scores, and the IoU threshold T   (For example, T = 0.5)   M: max selected boxes**Output:**   a group of bounding boxes that have been checked off**Algorithm: Steps to calculate NMS**1.     Arrange the bounding boxes based on their score2.     Repeat till there are no more boxes present:  (a)     Select the box with the best score. Name it $A_{s}$.  (b)     Remove the remaining boxes b using         IoU(b, $A_{s}$) ≥ T.


### 6.4. Bounding Box Regressor and RoI Classifier

Each RPN RoI that is received by this step generates two outputs. Bounding box optimization is similar to RPN in that it refines the position and dimensions of the bounding box and is employed by the network to classify regions into particular groups.

### 6.5. RoI Pooling

This RPN bounding box optimization step will crop and resize a specific area of the point chart. This can be accomplished by applying bi-linear interpolation and region of interest align (RoIAlign) on a feature map sampling point. The crop and resize feature of TensorFlow is used to perform this.

### 6.6. Segmentation Masks

This network, a convolutional network, chooses which positive zones to utilize as input for the creation of masks using the RoI classifier. Only 28 × 28 pixels make up the incredibly low resolution of this mask. The final masks are created by scaling down the ground-truth mask to 28 × 28 during the training phase and scaling up the predicted mask to the size of the RoI bounding box during the inference phase in order to calculate the loss.

#### 6.6.1. Loss Functions

In Mask-RCNN, a multithread loss function that incorporates segmentation, localization, and classification loss is used for each sampled RoI, as shown in Equation (Equation 8):(8)$$Lf=Lf_{cls}+Lf_{bbox}+Lf_{mask}$$
where: $Lf_{cls}$ = loss of classification; $Lf_{bbox}$ = loss of bounding box regression; $Lf_{mask}$ = loss of mask.

The total loss is obtained by taking the average of all losses across all samples.

#### 6.6.2. Classification Loss $Lf_{cls}$

The loss of RoI classification $Lf_{cls}$ is a logarithmic loss i.e., determined using Equation (Equation 9):(9)$$Lf_{cls}\left(p,s\right)=-log\left(p_{s}\right)$$
where: *s* = RoI true class; *p* = ($p_{0}$, …, $p_{k}$): predicted *k* + 1 class probability distribution.

#### 6.6.3. Bounding-Box Regression Loss $Lf_{bbox}$

Using Equation (Equation 10), the RoI bounding-box regression loss $Lf_{bbox}$ is determined as follows:(10)$$Lf_{bbox}\left(r^{u},t\right)=\sum_{i\in x,y,w,h}smooth_{L_{1}}\left(r_{i}^{u}-t_{i}\right)$$
where
$$smooth_{L_{1}}\left(z\right)=\left\{\begin{array}{cc} 0.5z^{2}, & if\;|z|<1\\ |z|-0.5, & otherwise \end{array}\right.$$

*s* = the RoI’s true class; *t* = $\left(t^{x},t^{y},t^{w},t^{h}\right)$ the RoI regression targets of an actual bounding-box; $r^{u}$ = $\left(r_{x}^{u},r_{y}^{u},r_{w}^{u},r_{h}^{u}\right)$ bounding-box regression predicted by class ‘*u*’.

#### 6.6.4. The Mask Loss $L_{mask}$

The mean binary cross-entropy loss or RoI mask loss $L_{mask}$ is calculated using Equation (Equation 11):(11)$$L_{mask}=\frac{-1}{m^{2}}\sum_{ij}\left[q_{ij}logp_{ij}^{s}+\left(1-q_{ij}\right)log\left(1-p_{ij}^{s}\right)\right]$$
where: *s* = the RoI’s true class; $q,p^{s}$ = masks for the class ‘*s*’ that is both true and expected in terms of RoI, respectively, $\left(q_{ij}\in \left\{0,1\right\},p_{ij}^{s}\in \left[0,1\right]\right)$.

With each class indicating the label of the actual mask and the expected value, each RoI is given a mask with a dimension of m × m.

### 6.7. Training Phase

The total number of images used for training, validation, and testing is 29,979 which is shown in Table 1. After including weights from the MS-COCO dataset in our model, we trained the network. Out of 29,979 total images, we used 2998 validation images and 17,987 training images to train our model leveraging stochastic gradient descent. The hyperparameters used for the model’s implementation are input image size 224 × 224, optimizer Adam, learning rate 0.001, batch size 128, loss categorical cross-entropy, epochs 20, and 100 training steps per epoch. These are used to fine-tune Mask-RCNN which is pretrained on MS COCO weights and the Bi-LSTM model.

### 6.8. Testing Phase

In the testing phase, we used 8994 images to test the trained model. In this testing data, each UAV image has a class label, masked segment, and bounding box that are predicted using the trained model. The predicted bounding boxes and labels should correspond to those in the dataset to evaluate the performance of the trained model for human activity recognition.

## 7. Bidirectional Long Short Term Memory (Bi-LSTM)

Bidirectional long short-term memory, often known as Bi-LSTM, is an LSTM model extension. Bi-LSTMs, unlike baseline LSTMs (which train a model in a single direction, i.e., forward, and only use past data), train a model in two ways, forward and backward, as seen in the following Figure 10. It uses two LSTMs, one for the forward process and another for the backward process. The following section provides a detailed explanation of how LSTM works. When the complete set of time-series data is accessible, the model learns a sequence of inputs from past to future in the forward direction, and from future to past in the backward direction. Because it executes processing in both directions, the calculation of the output frame at timestamp ‘*t*’ is dependent on the previous frame at a time ‘t−1’ and the next frame at a time ‘t+1’.

To preserve past and future information, this method employs two hidden states, one for the forward pass and the other for the backward pass. These states should be integrated to allow the network to produce more accurate predictions, and this method is known as merging. This can be accomplished using the sum, average, multiplication, and concatenation functions. Concatenation is the default technique for these functions. The following Algorithm 4 describes the Bi-LSTM procedure for human activity recognition.
**Algorithm 4** Procedure for Bi-LSTM model**Input:**$I_{no}\leftarrow$ Input layers count$H_{no}\leftarrow$ Hidden layers count$O_{no}\leftarrow$ Output layers count$S_{no}\leftarrow$ Data set instances count**Output:**Weights are associated with all of the inputs from all layers**Steps:**1.       **Forward pass:**    Run every input value for a single slice with 1 < = t < = T and determine all predicted results.    (a)    for i = 1 to $H_{no}$    (b)    for j = 1 to $S_{no}$ calculating the forward pass for the forward hidden layer’s activation function using the Equation (Equation 19) (from t = 1 to t = T)    (c)    end for    (d)   for j = $S_{no}$ to 1 calculating the backward pass for the backward hidden layer’s activation function using the Equation (Equation 19) (from t = T to t = 1)    (e)   end for    (f)   end for    (g)   for i = 1 to $O_{no}$ calculating the forward pass for the output layer using the previously stored activations using the Equation (Equation 20)    (h)   end for2.       **Backward pass:**    Calculate the portion of the objective function derivative for the forward-pass time slice with 1 <= t <= T.    (a)   for i = $O_{no}$ to 1 calculating the backward pass for the output layer using the previously stored activations using the Equation (Equation 20)    (b)   end for    (c)   for i = 1 to $H_{no}$    (d)   for j = 1 to $S_{no}$ calculating the backward pass for the forward hidden layer’s activation function using the Equation (Equation 19) (from t = T to t = 1)    (e)   end for    (f)   for j = $S_{no}$ to 1 calculating the forward pass for the backward hidden layer’s activation function using the Equation (Equation 19) (from t = 1 to t = T)    (g)  end for    (h)  end for3.       Update the weights of the network using each pass Equation (Equation 16).


### 7.1. LSTM Architecture

LSTM functions similarly to RNN, but it has one essential feature that distinguishes it from RNN: it saves information for future cell processing. The three gates of an LSTM cell are the forget gate, input gate, and output gate. The internal process of an LSTM cell is shown in Figure 11.

It has a memory pool with two key state vectors.
1.Short-term state: A hidden state is sometimes known as a short-term state. The output is kept at the current time step in this state. $S_{t-1}$ represents the preceding timestamp’s short-term state, while $S_{t-2}$ represents the current timestamp.2.Long-term state ($L_{t-1}$): A cell state is another name for a long-term state. This state examines and rejects data as it passes over a network that is intended for long-term storage. $L_{t-1}$ represents the preceding timestamp’s long-term state, while $L_{t}$ represents the current timestamp.

All timestamps and information are included in the cell state. The decision to read, write, or store is based on the activation functions whose outputs lie in between (0, 1), as shown in diagram Figure 11.

### 7.2. Forget Gate

This is the first state in the LSTM network’s cell. This gate determines whether the previous timestamp’s information should be stored or ignored. The forget gate, Equation (Equation 12), is as follows:(12)$$F_{t}=\sigma \left(C_{t}\times U_{f}+S_{t-1}\times W_{f}\right)$$
where: $C_{t}$ = The current timestamp *t* input; $U_{f}$ = The input’s weight; $S_{t-1}$ = The previous timestamp’s short-term state or hidden state; $W_{f}$ = The short-term state’s weight matrix.

The activation function, namely the sigmoid function, is then applied to it, yielding the value of $f_{t}$ in between (0, 1). The previous timestamp’s long-term state is then multiplied by it, as indicated in the computations below using formulae Equations (Equation 13) and (Equation 14).
(13)$$L_{t-1}\times f_{t}=0,\;if\;f_{t}=0$$
(14)$$L_{t-1}\times f_{t}=L_{t-1},\;if\;f_{t}=1$$

If the $f_{t}$ value is 0, everything is forgotten; otherwise, nothing is remembered.

### 7.3. Input Gate

This is used to manage the flow of input values into the cell and to quantify the importance of the most recent information. The following Equation (Equation 15) is the input gate equation:(15)$$I_{t}=\sigma \left(C_{t}\times U_{i}+S_{t-1}\times W_{i}\right)$$
where: $C_{t}$ = Current timestamp *t* input; $U_{i}$ = Matrix of input weights; $S_{t-1}$ = Previous timestamp’s short-term state or hidden state; $W_{i}$ = The weight matrix for the short-term state.

The activation function is then passed through the sigmoid function, yielding the value of ‘*I*’ at timestamp ‘*t*’. The value exists between (0, 1).

Latest information (or new information):(16)$$N_{t}=tanh\left(C_{t}\times U_{c}+S_{t-1}\times W_{c}\right)$$

This most recent information in Equation (Equation 16) is a function of the short-term state at timestamp ‘t−1’ and input ‘C’ at timestamp ‘*t*’. This data is required to obtain it through the long-term state. After applying the tanh activation function to it, the value of the most recent information falls between (−1 and 1).

This information is deleted from the long-term state if $N_{t}$ is negative, and it is added to the long-term state at the present timestamp if $N_{t}$ is positive. The following Equation (Equation 17), has been updated to include $N_{t}$ in the long-term state.
(17)$$L_{t}=F_{t}\times L_{t-1}+I_{t}\times N_{t}$$
where $L_{t-1}$ represents the long-term state at the current timestamp and others represent previously determined values.

### 7.4. Output Gate

This is utilized to determine the generation of the output from the current internal long-term state to the next short-term state and to govern the cell used for calculating the output activation of the LSTM unit. The output gate Equation (Equation 18) is as follows:(18)$$O_{t}=\sigma \left(C_{t}\times U_{o}+S_{t-1}\times W_{o}\right)$$

This formula is comparable to the forget and input gates. When the sigmoid activation function is applied to this equation, the output value is between 0 and 1. The current short-term state $O_{t}$ and tanh of the revised long-term state will then be computed using the following Equation (Equation 19):(19)$$H_{t}=O_{t}\times tanh\left(L_{t}\right)$$

That is, the short-term state is a function of current output and the long-term state is a function of tanh. Then, using the following Equation (Equation 20), apply the SoftMax activation function to the short-term state $L_{t}$ to obtain the output of the current timestamp. The prediction is the token with the highest score in the output.
(20)$$Output=SoftMax(S_{t})$$

## 8. Experimental Results

The dataset used to evaluate the models in this segment was first briefly described. Secondly, after integrating the HOG with Mask-RCNN for producing the human class, bounding boxes, and masks, we reported the classification results by employing the Bi-LSTM architecture. In the following part, we present the comparison findings of existing models with the suggested model. All of these tests are carried out on the Anaconda platform.

### 8.1. YouTube-Aerial Dataset

This new dataset was compiled by us using YouTube drone videos. This dataset contains activities that correlate to eight actions of the University of Central-101 (UCF101). BandMarching, Biking, CliffDiving, GolfSwing, HorseRiding, Kayaking, SkateBoarding, and Surfing are among the activities. The videos in this dataset feature varied heights for aerial filming, which involve big and quick camera movements. Below are a few samples of videos from this dataset. There are 50 videos for each action. The dataset division comprises 60%, 10%, and 30% of videos for training, validation, and testing, respectively, using the University of Central Florida’s aerial camera, rooftop camera, and ground camera (UCF-ARG) dataset. The YouTube-Aerial dataset is summarized in Table 1.

### 8.2. Metrics

In this section, we will see the metrics used to measure the performance of human detection and human action recognition as well.

#### 8.2.1. mAP (over IoU)

The effectiveness of the two-stage model, in this case, employed to detect humans, is determined using the mean average precision (mAP) over intersection over union (IoU) accuracy metric. The amount of overlap between two areas is assessed using the IoU metric. Using Equation (Equation 21), we determine IoU as the combined area divided by the total area of the two areas *S* and *T*:(21)IoU(S,T)=(S∩T)/(S∪T)

Consider Figure 12 for an example of how to compute the IoU:

#### 8.2.2. Accuracy

For this experiment, we employed the accuracy, precision, recall, and F1-score measures as performance evaluation measures to recognize human activities. The classification results obtained from testing the proposed model on the YouTube-Aerial dataset are shown in Figure 13 using the confusion matrix. Each cell gives the accuracy for each class metric in percentage.

The parameters accuracy, precision, recall, and F1-score are calculated by the following subsequent Equations (Equation 22)–(Equation 25). Where, the variables TP,TN,FP, and FN are denoted as true positive, true negative, false positive, and false negative.
(22)$$Accuracy=\frac{TP+TN}{TP+TN+FP+FN}$$
(23)$$Precision=\frac{TP}{TP+FN}$$
(24)$$Recall=\frac{TP}{TP+FN}$$
(25)$$F1-score=\frac{2\times Recall\times Precision}{Recall+Precision}$$

The accuracy was computed using the scores returned by the action classifier. Our proposed model outperforms previous models, resulting in an overall accuracy of 99.25%. The proposed deep learning-based human activity recognition model reached 99.25% classification accuracy, 99.23% precision, 99.25% recall, and 99.24% F1-scores. The following Figure 14a,b report the accuracy and loss of the proposed model (HOG + Mask-RCNN + Bi-LSTM).

### 8.3. Results

For the YouTube-Aerial dataset, experimental findings showed an accuracy of 99.25%. Mask-RCNN plays a key function in increasing the visibility of the activity and focusing on the human silhouette. Mask-RCNN is different from other implementations of this architecture, including RCNN and faster R-CNN, in that its output is a mask on the object rather than a bounding box, and this mask aids us in limiting the area of the object (in our experiment, a human). The established mask reduced the number of potential masks and highlighted the finished action, enabling us to provide useful inputs for the classification process. The following figures depict the outcome of implementing the first two stages of our proposed model. The output of the HOG feature descriptor algorithm is shown in Figure 15, which is fed into the Mask-RCNN.

Some of the experimental findings from utilizing the Mask-RCNN approach are shown in Figure 16, including the original and masked images, human images with masks and scores, and human segmented images. The following Figure 17 reports the classification results after extracting weights from the Bi-LSTM network which are helpful for recognizing the activities of a human.

#### 8.3.1. Comparison with State-of-the-Art Methods

The human detection model (i.e., HOG with Mask-RCNN) was trained to recognize humans using a YouTube-Aerial dataset that contained human activity video sequences which are divided as sequences of frames so that it could recognize and label humans in frames. This model had an mAP of 99.55% for this dataset after being trained and evaluated with the validation set. Below, Table 2 summarizes HOG and Mask-RCNN performance to other models in terms of mAP.

The human detection model was compared with other object detection models in Figure 18, however, several datasets were used to train the models. The study has enhanced its performance by comparing the model with the other existing models on the same dataset. The following Figure 17 and Figure 19 show a visual comparison that includes the results from other models on the same dataset (i.e., YouTube-Aerial dataset), where our model succeeded and other models failed in terms of classification results.

We now compare our suggested approach to cutting-edge methods employed for human detection and tracking. Our primary goal is to find and follow humans in environments with known challenges. In ways to construct more challenging human activity imagery in the training data, we combined the HOG and Mask-RCNN techniques. This allows the deep network to study human activity detection features in unrestricted environments. For human detection and tracking, several experiments have been conducted, each with a distinct classification and accuracy rate.

A comparison between the suggested method with cutting-edge techniques is summarized in Table 3. Below Figure 20 displays the comparison chart of the proposed human activity detection model with the human detection and tracking models.

#### 8.3.2. Discussion

As described in Table 3, fast R-CNN, which is scale-aware and capable of recognizing people across a range of distances, was used by the authors in [79]. The suggested framework unifies small-scale and large-scale subnetworks into a single design. After integrating the outputs of each subnetwork, detection results are produced. A single-shot detector (SSD) approach for detecting pedestrians was given by the authors in [81]. The SSD convolutional neural network first collects shallow features, and they are combined with the convolutional layer of deep semantic data. The individual is eventually found in the still pictures. The proposed method employs preselection boxes with various ratios that enhance the overall model’s capacity for detection. Using the region proposal network and other data, the authors in [77] provide a unique baseline. The computation of the convolutional features map and bounding boxes was the primary goal of the region proposal network in the suggested pipeline. In order to categorize the characteristics extracted by the region proposal network, a cascaded boosted forest was used. It appears that the plan utilized in this model produced useful results. For person detection and tracking, the authors of [13] presented a hybrid technique based on CNN and V-disparity. The region of interest is extracted using the V-disparity approach, and the input is supplied to the CNN architecture.

Convolutional neural networks and recursive neural networks are integrated with the proposed work to efficiently analyze the obtained features, then classify them. To address the issues with human detection, the authors of [78] suggested a faster R-CNN based solution in conjunction with skip pooling. The design of faster R-CNN’s proposal network is expanded to a multi-layer framework and then integrated with skip pooling. Without considering the intervening layers, skip-pooling networks take several RoIs from lower layers and feed them to higher layers.

The study in [80] offered a compression approach for carrying out human detection tasks that were built upon the teacher-student framework and standard random forest (RF). By adopting the soft version of the teacher’s RF output, the student’s shallow RF in the compression network is educated by imitating the performance of the teacher’s shallow RF. The accuracy of the human detection network utilized by the authors in [84] was increased by a total of 17%. For quick human detection in real-time, Xu et al. [82] suggested an enhanced mask R-CNN model that achieved an accuracy of 88%. A noteworthy model based on a two-stream unified deep network was presented by Wang et al. [83] in order to detect humans with more accuracy and with less computational effort.

Furthermore, the HIT (human image-threshing) machine proposed in [85] detects the human body using Mask-RCNN, image cropping, and resizing using a facial image threshing (FIT) machine, and activity classification using a deep learning model. This model used the HAR dataset to recognize activities using the smartphone camera, stretch sensor, and inertial measurement unit (IMU). If HAR systems use cameras as their input source, HIT machine-based HAR systems are a good healthcare alternative. It attained 98.53% accuracy when the ResNet architecture served as its deep learning model. Another model [86] combined multi-channel attention networks using transfer learning to presume a convolutional neural architecture for human activity recognition in still images. Four CNN branches were employed in this model to create feature fusion-based ensembling, and each branch contained an attention module that was used to extract contextual data from the feature map created by previously existing pretrained models. In order to obtain the final recognition output, the derived feature maps from the four branches were combined and fed into a fully connected network. This model evaluated the system using three different datasets: the Willow human activities dataset, the BU-101 dataset, and the Stanford 40 actions dataset. A unique HAR system with good performance was suggested in another model by combining EfficientDetD7 for detecting humans, EfficientNetB7 for extracting the feature, and LSTM for the classification of time series data [87]. By introducing numerous distortions, such as blur, noise, and illumination variations, this research added new challenges to the UCF-ARG aerial dataset. The result is a reliable HAR system that integrates EfficientDetD7, EfficientNetB7, and LSTM for activity classification and human detection.

Using the proposed model, all activities were correctly identified with a greater classification performance because the Bi-LSTM’s ability to use past and future observations allows for correct differentiation of dynamic human motions and reported an accuracy of 99.25%. Bidirectional LSTM architecture is used to extract the temporal features from the video. The model is used to recognize activities on a pool of eight activities, in this case, that are closely related to one another, meaning that they either have a similar data sequence or visually resemble one another. Activities such as kayaking and surfing are slightly unclassified for some frames in the video, with considerably less error rate.

## 9. Conclusions

In this research, the proposed deep learning action recognition framework sequentially leverages the temporal and spatial features in the end-to-end network. The network is comprised of four primary components. Initially, the input frames are given to HOG to identify the patterns in image data and extract them. Next, these features are refined through the pretrained Mask-RCNN for extracting the visual features. The bi-directional LSTM network processes the refined feature maps by exploiting the temporal relationship between the frames for the underlying action in the scene. After that, a fully connected layer with a softmax classifier following the bi-direction LSTM is employed to assign the probabilities to the actions of the subject in the scene. The marginal loss function is used in conjunction with cross-entropy loss to penalize the network for the inter-action class while also improving the network for variations in intra-action. The YouTube-Aerial dataset is deployed to train and validate the network, and a total of eight actions are addressed. The network is assessed using common performance metrics such as IoU and mAP. The quantitative findings demonstrate encouraging validation test outcomes, and attained an accuracy of roughly 99.25%. In the future, we intend to test our system on a variety of video sequences, including those captured by various other multirotor drones, scenarios from aerial feeds, and simulation clips shot from differing viewpoints. Additionally, we also intend to optimize our framework so that it can function on low-end hardware and yield significant real-time analytics.

## Figures and Tables

**Figure 1 sensors-23-02569-f001:**
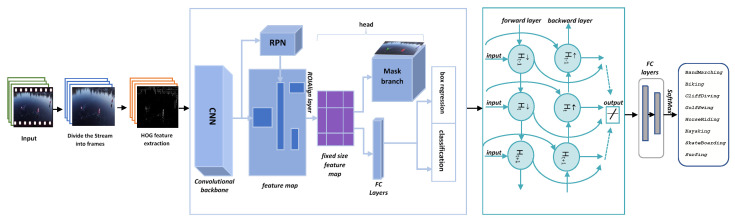
Human activity recognition in UAV videos using HOG, Mask-RCNN, and Bi-LSTM architectures.

**Figure 2 sensors-23-02569-f002:**
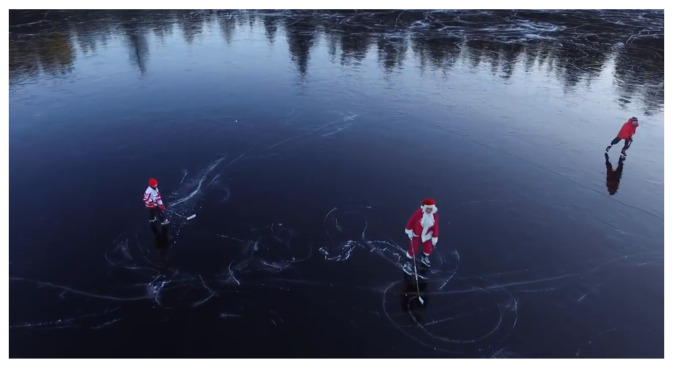
Input frame.

**Figure 3 sensors-23-02569-f003:**
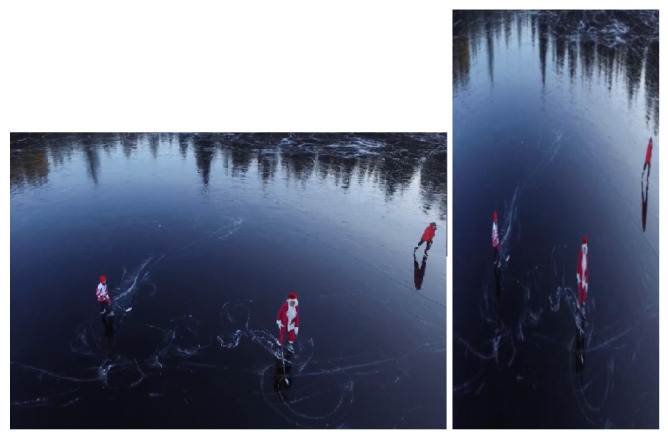
Actual frame and rescaled frame.

**Figure 4 sensors-23-02569-f004:**
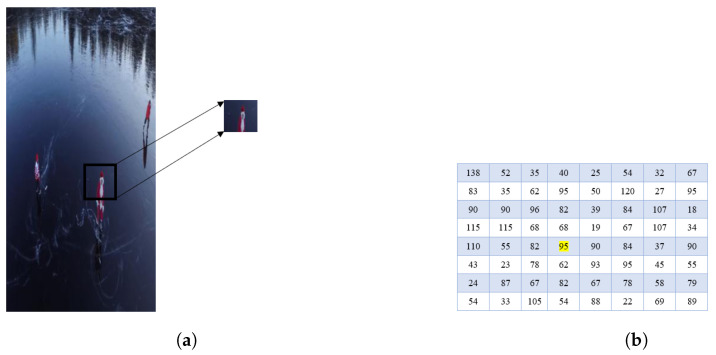
Visualization of extracted small patch and pixel matrix of the chosen patch. (**a**) Abstract a small patch. (**b**) Pixel matrix.

**Figure 5 sensors-23-02569-f005:**
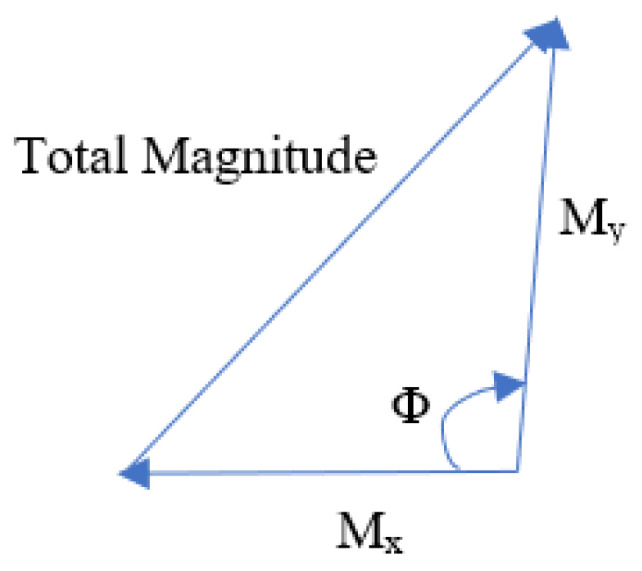
Pythagoras theorem.

**Figure 6 sensors-23-02569-f006:**
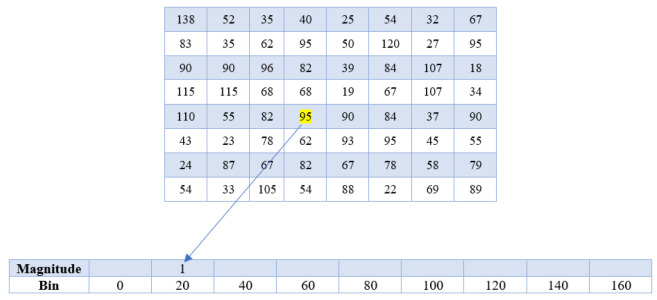
Mapping pixels to bins.

**Figure 7 sensors-23-02569-f007:**
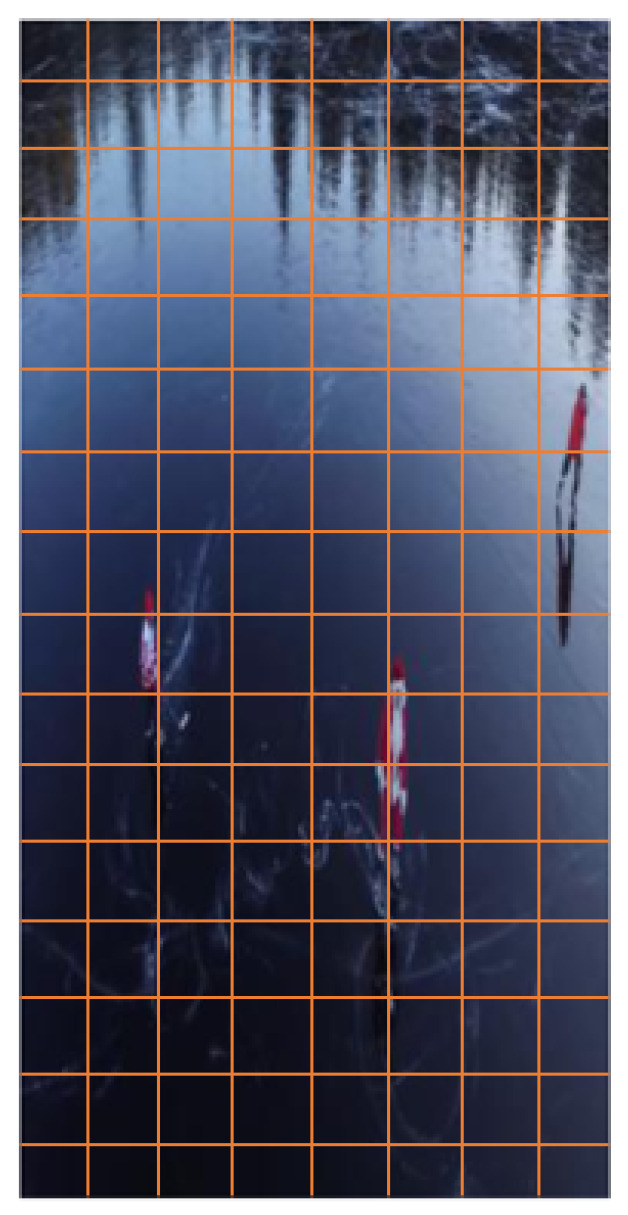
Dividing the frame into cells.

**Figure 8 sensors-23-02569-f008:**
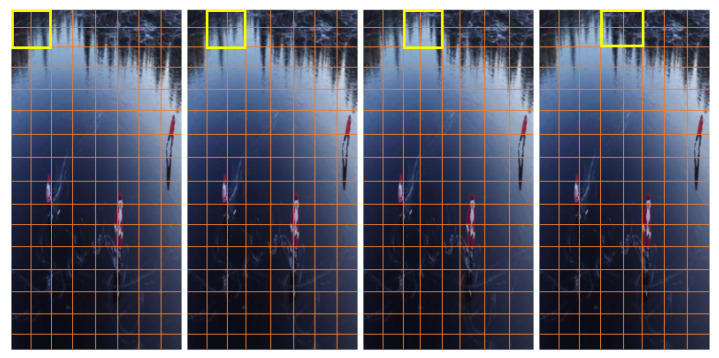
Creating blocks.

**Figure 9 sensors-23-02569-f009:**
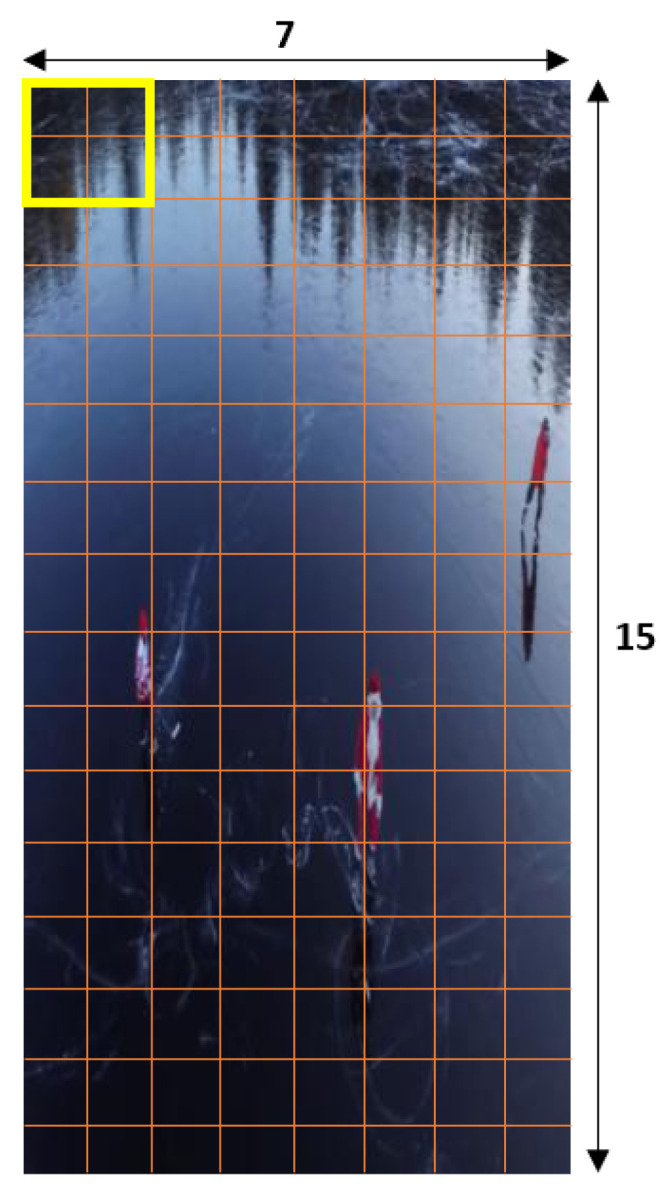
HOG features generation.

**Figure 10 sensors-23-02569-f010:**
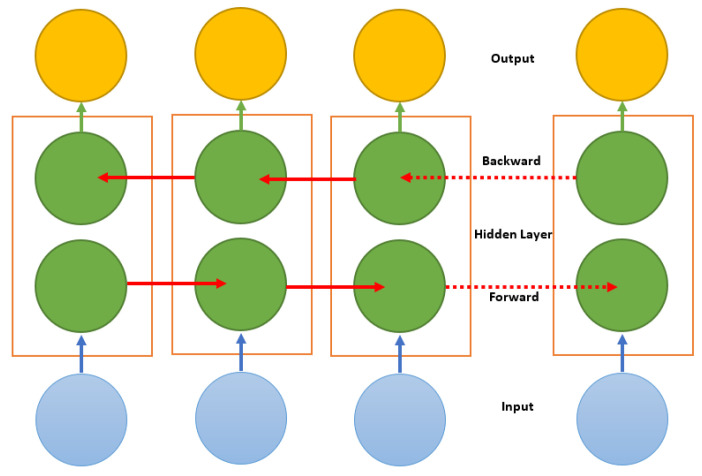
Bidirectional long short-term memory architecture.

**Figure 11 sensors-23-02569-f011:**
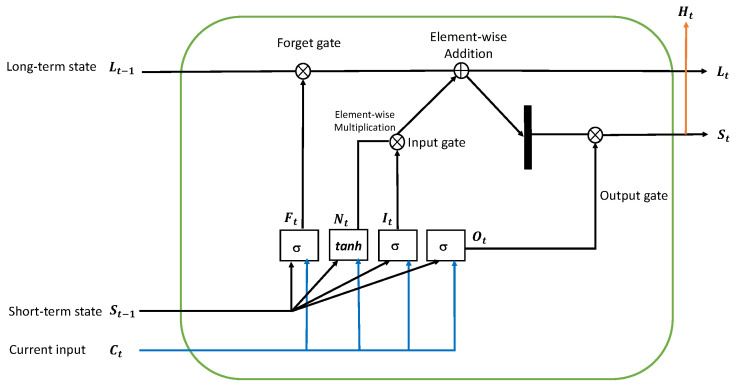
Long short-term memory architecture.

**Figure 12 sensors-23-02569-f012:**
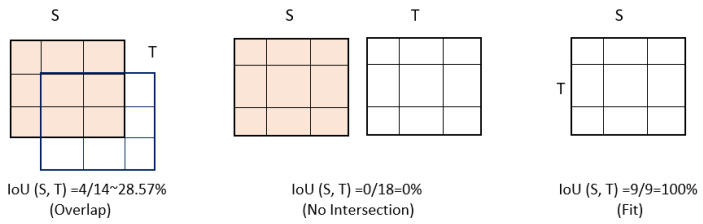
Calculation of IoUs.

**Figure 13 sensors-23-02569-f013:**
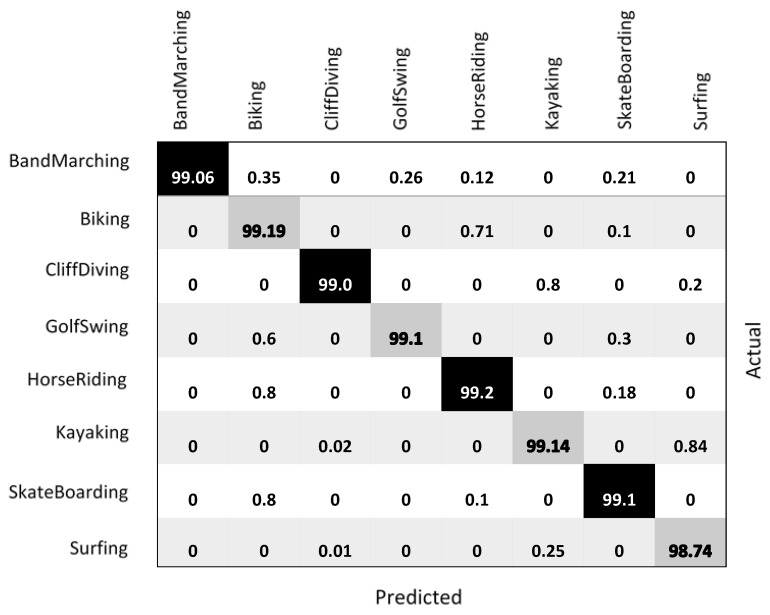
Classification results evaluated from testing the proposed model on the YouTube-Aerial dataset as a confusion matrix. Each cell gives the accuracy for each class measure in percentage.

**Figure 14 sensors-23-02569-f014:**
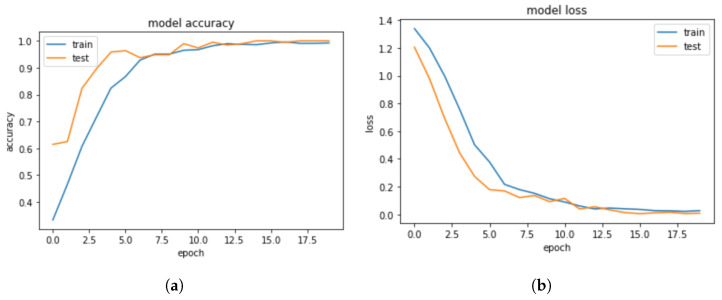
Proposed model accuracy and loss. (**a**) Model accuracy. (**b**) Model loss.

**Figure 15 sensors-23-02569-f015:**
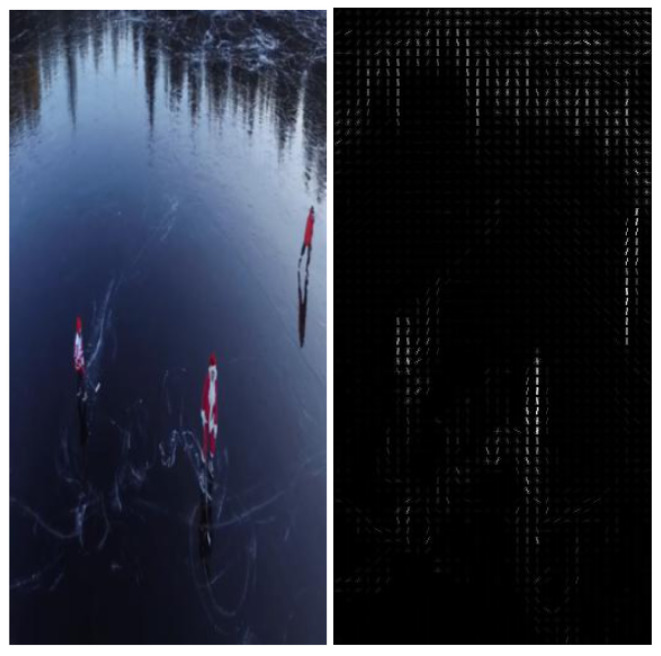
Resized frame and HOG output.

**Figure 16 sensors-23-02569-f016:**
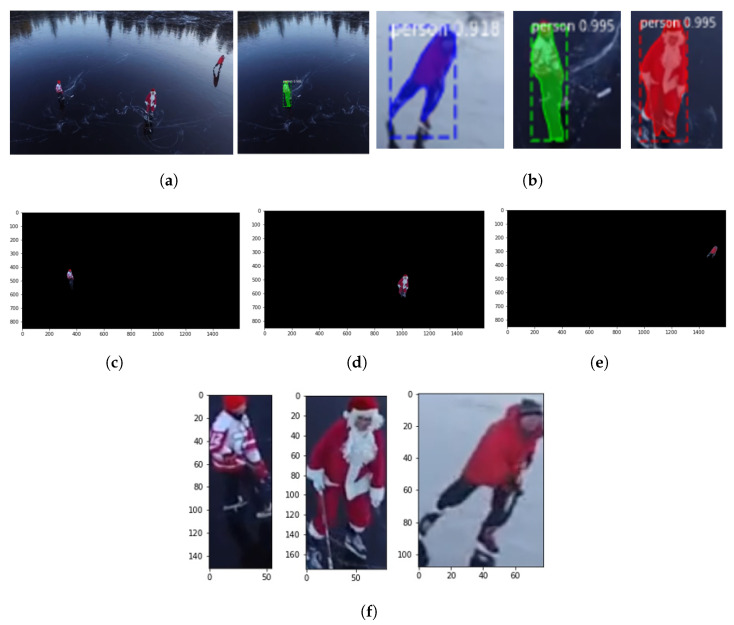
Outcomes of human segmentation using Mask-RCNN. (**a**) Original frame and masked frame. (**b**) Human images with masks and scores. (**c**) First human. (**d**) Second human. (**e**) Third human. (**f**) Segmented humans.

**Figure 17 sensors-23-02569-f017:**
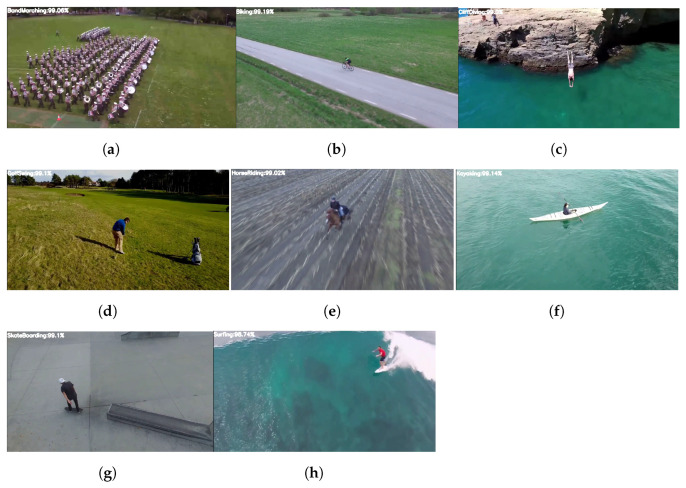
Human activities with the percentage of accuracies. (**a**) BandMarching. (**b**) Biking. (**c**) CliffDiving. (**d**) GolfSwing. (**e**) HorseRiding. (**f**) Kayaking. (**g**) SkateBoarding. (**h**) Surfing.

**Figure 18 sensors-23-02569-f018:**
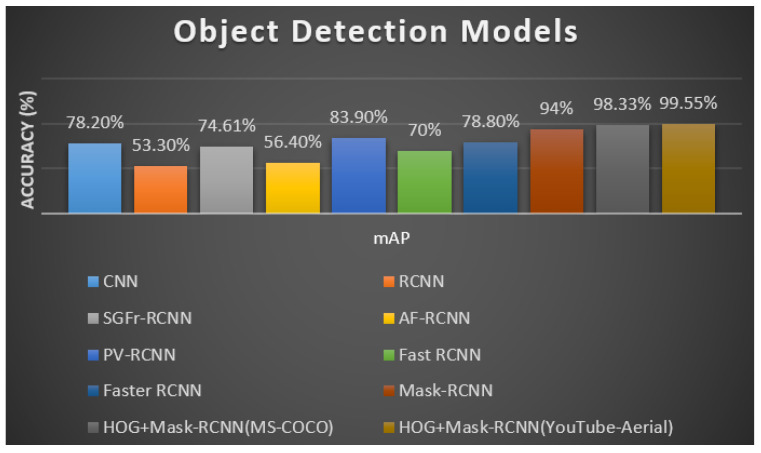
Bar graph of object detection models.

**Figure 19 sensors-23-02569-f019:**
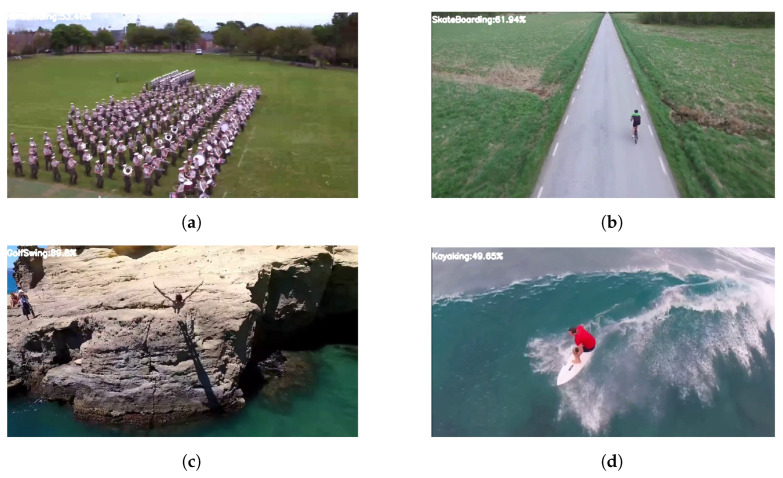
Some of the unclassified results from the other state-of-the-art models. (**a**) BandMarching unclassified as HorseRiding. (**b**) Biking unclassified as SkateBoarding. (**c**) CliffDiving unclassified as GolfSwing. (**d**) Surfing unclassified as Kayaking.

**Figure 20 sensors-23-02569-f020:**
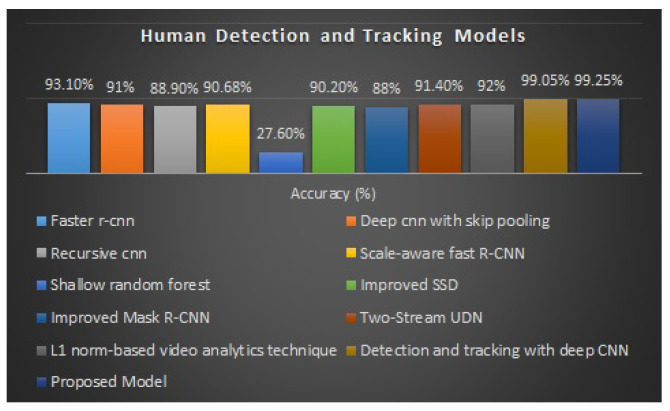
Bar graph of human detection and tracking models.

**Table 1 sensors-23-02569-t001:** Summary of YouTube-Aerial dataset.

Feature	Value
Actions	8
Clips per class	50
Total clips	400
Total frames	29,979
Clip length	max 97 Sec

**Table 2 sensors-23-02569-t002:** Different object detection models with the percentage of mAP.

Object Detection Models	mAP
CNN [47]	78.2%
RCNN [71]	53.3%
SGFr-RCNN [72]	74.6055%
AF-RCNN [73]	56.4%
PV-RCNN [74]	83.9%
Fast RCNN [75]	70%
Faster RCNN [45]	78.8%
Mask-RCNN [12]	94%
HOG + Mask-RCNN (MS-COCO) [76]	98.33%
HOG + Mask-RCNN (YouTube-Aerial)	99.55%

**Table 3 sensors-23-02569-t003:** Different human detection and tracking models with accuracies.

Human Detection and Tracking Models	Accuracy (%)
Faster R-CNN [77]	93.1%
Deep CNN with skip pooling [78]	91%
Recursive CNN [13]	88.9%
Scale-aware fast R-CNN [79]	90.68%
Shallow random forest [80]	27.6%
Improved SSD [81]	90.2%
Improved mask R-CNN [82]	88%
Two-stream UDN [83]	91.4%
L1 norm-based video analytics technique [84]	92%
Detection and tracking with deep CNN [14]	99.05%
Proposed model	99.25%

## Data Availability

Not applicable.

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
