# Peer review of "Vision-Based HAR in UAV Videos Using Histograms and Deep Learning Techniques"

_sensors, 2023, doi:10.3390/s23052569_

Round 1

Reviewer 1 Report

The paper proposes an interesting approach for detection from UAV videos. However, it requires several revisions before publication. Below are my specific comments. 

1. Abstract is problematic in terms of English and does not include quantitative research findings.

2. Introduction lacks proper referencing. Several paragraphs of information without citation. 

3. Authors did not discuss the effect of rescaling process in Section 5.2. There is an obvious distortion of objects as can be seen in Figure 3 and 4 etc.

4. Figure 17 replicates Table 3 and also gray-level coloring of the bars makes it difficult to understand. So FÄ°gure 17 removal is advised.

5. As Authors compared their architecture with several object detection models, a figure which also includes the results from other models is required (a visual comparison where your model succeed and others failed, etc).

6. Possible limitations and future research possibilities should be discussed. Actually paper needs a comprehensive and separate discussion section.

7. Conclusion should be rewritten to reflect what significant findings are achieved compared to state-of-the-art and provide quantitative results. 

Reviewer 2 Report

Dear Authors,

Please find the attached file for your reference. Please update the paper based on the comments and resubmit it.

Regards

Round 2

Reviewer 1 Report

The revised version satisfies all the concerns that I raised in previous version. Revisions are solid and improved the quality of the paper. I recommend a full check of English

Reviewer 2 Report

Dear Authors,

Thank you for addressing all my comments and I don't have any further concerns.

Regards 

Author Response

Dear Reviewer,

The authors would like to thank the reviewer for his extraordinary efforts in terms of detailed review and constructive comments. With all his comments and suggestions, the quality of the paper has been enhanced a lot. Once again thanks for approving our revisions for the publication of the article.